# Consumers' Perception of In-Vitro Meat in New Zealand Using the Theory of Planned Behaviour Model

Maya Murthy Malavalli [1], Nazimah Hamid [1,*] , Kevin Kantono [1,2] , Ye Liu [1] and Ali Seyfoddin [3]

1 Department of Food Science, Auckland University of Technology, Auckland 1142, New Zealand; maya.malavalli@aut.ac.nz (M.M.M.); kkantono@aut.ac.nz (K.K.); ye.liu@aut.ac.nz (Y.L.)
2 Arla Innovation Centre, Arla Foods amba, N 8200 Aarhus, Denmark
3 Drug Delivery Research Group, School of Science, Auckland University of Technology, Auckland 1142, New Zealand; ali.seyfoddin@aut.ac.nz
* Correspondence: nazimah.hamid@aut.ac.nz

**Abstract:** The purpose of this study was to investigate the perception of in-vitro meat (IVM) among New Zealand consumers and to understand their purchase and consumption behaviour using the Theory of Planned Behaviour framework developed in this study. An online survey questionnaire was created using the Qualtrics software to understand the perception of IVM, based on the conceptual framework. Participants ($n = 206$) were recruited in this survey, and the data collected were subjected to PLS-PM analysis. The conceptual framework was tested for validity, and Goodness of fit (GoF). The internal validity was assessed using Cronbach's alpha, KMO value, inter-item correlation values (β-coefficients) and $p$-values. The findings suggest that variables such as environment and sustainability, health and safety, as well as current purchase and consumption behaviour have a strong relationship and a robust effect on IVM purchase and consumption behaviour. Consumers' cultural beliefs had minimal influence on IVM purchase likelihood. Results in this study also indicated that most New Zealand consumers had neutral opinions in terms of engaging with IVM.

**Keywords:** consumer perception; in-vitro meat; New Zealand; planned behaviour model

## 1. Introduction

In-vitro meat is a type of meat that is produced using animal cells, under laboratory conditions [1]. In-vitro meat (IVM) works on the principle of cellular agriculture where stem cells are extracted from donor animals through biopsy. A recent review by Post etal. (2020) reported on the scientific, sustainability and regulatory challenges of IVM [2]). The extracted cells usually belong to embryonic stem cells (ESC) [3], adult stem cells (ASC) [4,5], mesenchymal stem cells (MSC) [6,7] or induced pluripotent stem cells (iPSC's) [4]. These cells are cultured either by scaffolds or self-organising techniques [4] or are cultured in a sterile bioreactor. The extracted cells, under favourable conditions, undergo cell proliferation and differentiation to form myofibers, which eventually form muscle tissues. These muscle tissues combine to form skeletal muscles, which can be harvested as edible meat.

There has been extensive research on consumer perception of IVM in the past few years. Most studies on IVM perception have focused on consumers in the USA [8–11] and European countries [12,13]. There have however been a few studies on IVM with consumers from China and India [10], New Zealand [14], Australia [15], and Brazil and the Dominican Republic [16]. Consumer perception on IVM was generally found to be positive among European consumers. European consumers were more interested in engaging with IVM, with 52% of the Netherlands consumers definitely willing to try IVM [17], 68% of the United Kingdom consumers willing to eat IVM [18], and 44% of Italian consumers willing to buy IVM [19]. Acceptance of IVM by Europeans consumers [12,17,19–22] was due to increased IVM familiarity brought about by the unveiling of the first lab-grown burger in London, 2013. A study that investigated the perception of IVM in the USA using an online

survey [8] reported that nearly two thirds of the participants surveyed (*n* = 673) indicated that they would probably or definitely try IVM. However, only one third of the participants were definitely or probably willing to eat IVM regularly or as a substitute for farmed meat. This was attributed to concerns regarding the anticipated high price, limited taste and appeal, and the product being unnatural. It was concluded that although USA consumers were likely to try IVM, few believed that it would replace farmed meat in their diet. It is important to note that the various studies published in the literature utilised different descriptions of IVM, methodology, and sampling techniques. Hence a generalisation and direct comparisons of results should be made with caution. Another study [23] indicated higher acceptance of IVM with US consumers compared to UK consumers.

There are concerns about IVM in terms of taste [8], unnaturalness [8,13,24–26] disgust [13,27–29], price [19], perceived health and safety risk, and technology involved in IVM production [13]. IVM acceptance on the other hand is influenced by environmental [16] and animal welfare issues [12], ethical reasons [8], and perceived health benefits [13]. Literature has shown that consumers' perceptions on IVM has a strong effect on IVM purchase and consumption behaviour. In fact, positive perceptions on IVM have been linked to higher purchase likelihood. For example, consumers who believe that IVM is beneficial for the environment and animal welfare, are more likely to buy IVM [8,16,20] compared to those who think IVM is a waste of resources [30]. Research has shown that IVM purchase intentions are influenced by many factors such as consumers' environmental and welfare awareness [8], their openness towards sustainable alternatives [10], their awareness of IVM and its sustainability [19], IVM sensory qualities [12,19], and IVM health and safety [12,19]. Consumers with higher environmental and welfare awareness tend to have higher IVM purchase likelihood, due to the environmental and animal-friendly nature of IVM. Consequently, these consumers have higher purchase intentions compared to others [10]. Hence consumers with higher environmental and welfare awareness had higher openness towards sustainable alternatives (plant-based alternatives or animal-based).

Consumers with higher awareness on IVM and its potential benefits find IVM more acceptable and tend to have a higher purchase intention towards it. This was confirmed by many researchers [11,17,22] who indicated that purchase intentions are linked to many factors. Consumers who have positive perception on sustainable alternatives and agree that IVM is a sustainable alternative tend to have higher purchase intentions [10,19], as opposed to the pro-traditional meat individuals [12,30]. Consumers with strong sensory preferences, and health and safety concerns tend to have reduced purchase intentions towards IVM [20,30]. Consumers with higher meat consumption tended to have lower inclinations to IVM [14]. It is important to note that in these studies consumers did not consume and evaluate IVM meat. It was only their overall perception of IVM that was measured rather than their actual experience of consuming IVM.

The commercial success of IVM strongly depends on consumers' perception. In fact, consumer perception determines whether IVM will be acceptable or rejected. Therefore, it is essential to understand consumer perception of IVM. After the USA, meat consumption in New Zealand is one of the highest the world with consumption reported at 11.15 kg/capita [31]. In addition, NZ is economically invested in the traditional meat industry. Hence this study was carried out to understand the perception of IVM by New Zealand consumers, in terms of their willingness to try/buy IVM, and their reasons for acceptance/rejection of IVM. These findings will add valuable insights for future IVM markets. Furthermore, the information gained through this study will also help in creating better marketing strategies for companies producing IVM in the future.

Based on the studies that described consumers' attitudes related to such areas as the environment, sustainability awareness, health and safety perception, cultural beliefs, and their current meat purchase and consumption behaviour; the hypotheses of this research are as follows:

**Hypothesis 1.** *Consumers' views on the environment and sustainability have a significant effect on IVM purchase and consumption behaviour.*

**Hypothesis 2.** *Consumers' views on health and safety have a significant effect on IVM purchase and consumption behaviour.*

**Hypothesis 3.** *Consumers' cultural beliefs have a significant effect on IVM purchase and consumption behaviour.*

**Hypothesis 4.** *Consumers' current purchase and consumption behaviour have a significant effect on IVM purchase and consumption behaviour.*

## 2. Materials and Methods

*2.1. Ethics Statement*

This study was approved by the Auckland University of Technology Ethics Committee (AUTEC 19/68). Participants provided written and informed consent prior to commencement of the study.

*2.2. Research Design*

In order to explore the perception of IVM by New Zealand consumers, quantitative research was conducted using an online survey as it is a research tool that is widely used in consumer science studies, and a useful platform for collecting large amounts of anonymous responses. This study also attempts to measure behavioural intentions such as willingness to try (WTT) and willingness to buy (WTB), since the commercial success of IVM, as a sustainable alternative, lies in the WTT, WTB, and willingness to eat (WTE) IVM. These approaches were used with the USA, European and Asian consumers to determine their WTT, WTB and WTE IVM [8,10,19,32].

*2.3. Procedure*

2.3.1. Online Survey Recruitment

Consumer views were collected through an online survey in New Zealand, and the survey was administered using the Qualtrics Software (Provo, UT, USA). The survey was open to all New Zealanders aged between 18 to 65 years old, irrespective of their eating habits. The survey was administered in English, from March to September 2019. Most of the participants were recruited in-person at public spaces located in central Auckland, whilst the rest were recruited online through an advertisement posted on social networking sites like Facebook, Instagram and LinkedIn.

Participation in the research was entirely anonymous, voluntary, and confidential. Participants were under no obligation to take part in the study and had the freedom to withdraw at any stage without any further questions. The participants were given a coffee voucher as a token of appreciation. The eligibility criteria to take part in the study were respondents over the age of 18 and they must be living in New Zealand. A total of 265 participants were recruited, but once incomplete surveys were omitted, 206 completed surveys were obtained for further analyses. Participants were provided with the following information prior to administering the survey:

"With the increase in population, the demand for food has also increased. FAO experts suggest that by 2050, there will be a 100% increase in the demand for food. The demand for meat in particular will be about 73% higher. Rearing of ruminants such as cattle, sheep and goat are carried out on green pastures that are cleared by deforestation. Animal breeding has been employed to further improve and increase calf production (95% of cattle and 5% of beef cattle).

In the traditional meat industry animals are caged, castrated, confined to cages, and treated with antibiotics, pesticides and growth hormones, prior to being slaughtered. Most animal welfare associations are concerned about the well-

being of animals and seek to prevent unnecessary animal suffering and death. The negative environmental effects associated with meat production include pollution through use of fossil fuel, animal methane, effluent waste, and water and land consumption.

In order to address the growing environmental and ethical concerns amidst the negativity associated with production and consumption of meat, the production and perception of in vitro meat (IVM) has been increasingly researched. IVM production involves the culturing of stem cells from farm animals in bioreactors by employing advanced tissue engineering techniques. IVM advantages include being environmentally friendly, requiring lower energy consumption, lowering greenhouse gas emission, lowering land and water consumption, and resulting in low carbon footprint. In addition, IVM is high in protein, low in unhealthy fats, highly sustainable, environmentally friendly, ethical and animal friendly. Hence, IVM is quickly becoming the best sustainable alternative to conventional meat."

### 2.3.2. Online Survey Procedure

The questionnaire was mostly accessed either by scanning the QR code that contains an anonymous link generated by Qualtrics. The QR code was shared through links on Facebook, Instagram, and LinkedIn. Participants used a checkbox to indicate their consent to take part in the study, before answering the survey questionnaire. Before commencement of the survey, all participants were briefed about the study and were given some information (the information pack contained general information on annual worldwide demand for food, livestock farming practices, animal welfare, and advantages of consuming IVM) on in-vitro meat to educate, rather than influence consumers. Provision of information has been proven to improve the consumer perception [13,22,33] and also result in increased WTT IVM [17,22].

Participants were first asked demographical questions, followed by questions on eating behaviour and meat consumption. In addition, questions on participants' familiarity, as well as their willingness to try/buy IVM if commercially available, were also included. Participants were asked to rate their answers on a 5-point Likert scale. Furthermore, the questionnaire also included questions on other aspects of IVM such as marketing, regulatory, ethical, and religious views that indirectly affect the perception.

The data from the survey were automatically saved on the Qualtrics website and were extracted on completion of data collection. The collected raw data were analysed using XLSTAT (version 2018.5) (Addinsoft, New York, NY, USA) and R Studio (version 1.1463) operating with R-version 3.6.1.

### 2.3.3. Data Analysis

The conceptual framework model as shown in Figure 1 used in this study was developed to understand the effect of New Zealand consumer perception of IVM purchase and consumption behaviour. As stated in the hypotheses, consumers' views on environment and sustainability, health and safety, cultural beliefs and consumers' current purchase and consumption behaviour of meat play an important role in purchase intention of IVM. Consequently, most studies in the IVM literature examined the consumer perception of IVM and its effects on purchase likelihood [8,10–13,19,22,25,32–34]. However, the framing of the research objectives, hypothesis and variables examined vary with each study.

Previous studies investigating consumer perception of IVM used other regression models such as logistic regression, mixed logic and multiple regression models [8,10,16,22,34] and other classical statistical approaches such as ANOVA [11,16]. PLS-PM can be classified as a correlational structural equation modelling, which allows the estimation of complex cause and effect modelling through the use of latent and manifest variables. In this study, PLS-PM was used since it was a more suitable method for studying the complex multivariate relationships between latent and observed variables [35].

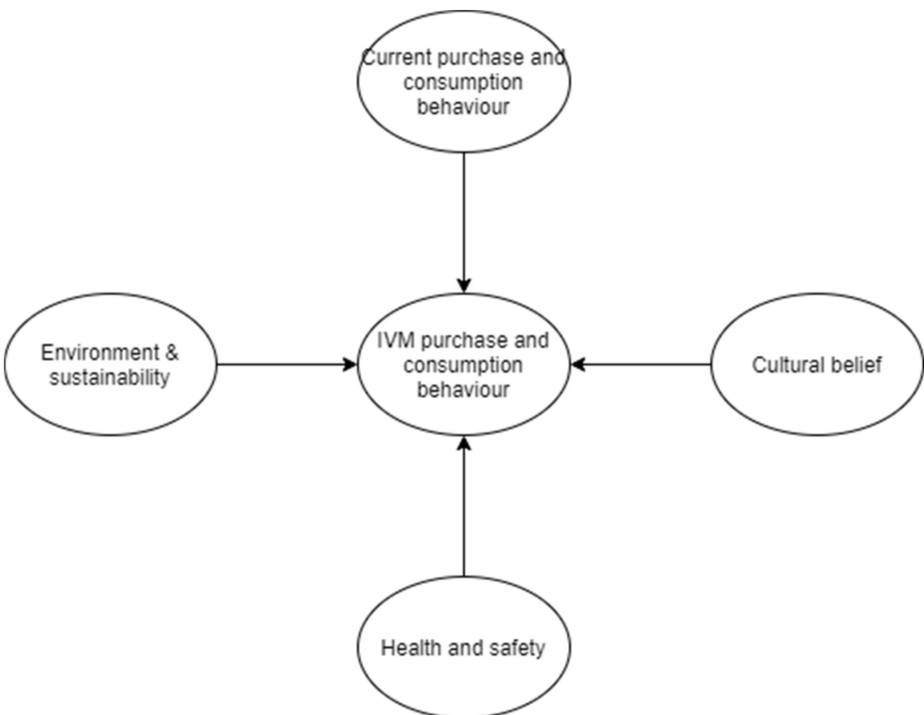

**Figure 1.** Conceptual model used in our study to test the hypotheses.

The conceptual model was validated and measured using Goodness of Fit statistics, and provided the overall model fit for PLS-PM. The internal validity of the variables was assessed using Cronbach's α and Kaiser Meyer Olkin (KMO) values. The Cronbach's alpha value measures the internal consistency and scale reliability, whereas KMO values measure the sampling adequacy for each variable and for the model. The above statistical tests are essential for justifying results and confirming the conceptual model. However, at this point, no inter-item correlation was performed due to the numerous variables involved in this study.

Since PLS-PM does not hold any distribution assumption [36], the resampling task was performed to validate and obtain further data on the variability of parameter estimates [37]. Accuracy of the PLS-PM model was obtained by bootstrapping. This is a non-parametric procedure that measures the significance of the PLS results such as coefficients, Cronbach's alpha, HTMT and the $R^2$ value [38,39]. The $R^2$ value of the conceptual model in this study was found to be 0.777. PLS-PM analysis was carried out using the PLSPM package version 0.4.9 [35] by R Studio (RStudio, Boston, MA, USA) (version 1.1463) operating with R-3.5.1.

## 3. Results and Discussion

### 3.1. Perception of IVM

#### 3.1.1. Sociodemographic Characteristics

Sociodemographic characteristics of the sample population involved in this study are summarized in Table 1. The demographic factors included gender, age, qualification, income, nationality, religious, eating habits and frequency of meat consumption. Most of the respondents belonged to the 18–25 years age group (73.79%), with more women participants (61.17%), meat-eaters/non-vegetarians (69%), and participants who consumed meat most of the time (48.54%).

**Table 1.** Sociodemographic characteristics of the sample by number and percentage of respondents.

| Variable | Category | Number | Percentage |
|---|---|---|---|
| Age | 18–25 | 152 | 73.79% |
| | 26–35 | 33 | 16.02% |
| | 36–45 | 11 | 5.34% |
| | 46–55 | 5 | 2.43% |
| | 56–65 | 4 | 1.94% |
| | Prefer not to say | 1 | 0.49% |
| Gender | Male | 77 | 37.38% |
| | Female | 126 | 61.17% |
| | Prefer not to say | 3 | 1.46% |
| Qualification | High School | 80 | 38.83% |
| | Diploma or certification | 27 | 13.11% |
| | Bachelor's degree | 56 | 27.18% |
| | Master's degree | 22 | 10.68% |
| | Postgraduate degree | 19 | 9.22% |
| | Prefer not to say | 2 | 0.97% |
| Income | <$20,000 per year | 62 | 30% |
| | Approx. $50,000 per year | 19 | 9% |
| | $50,000–$70,000 per year | 16 | 8% |
| | $70,000–$90,000 per year | 22 | 11% |
| | $90,000–$120,000 per year | 14 | 7% |
| | >$120,000 per year | 18 | 9% |
| | Prefer not to say | 55 | 27% |
| Respondents nationality | New Zealander | 59 | 29% |
| | Indian | 21 | 10% |
| | Chinese | 21 | 10% |
| | Other nationalities | 78 | 38% |
| | Prefer not to say | 27 | 13% |
| Religious | Yes | 90 | 43.69% |
| | No | 96 | 46.60% |
| | Prefer not to say | 20 | 9.71% |
| Eating habits | Vegetarian | 15 | 7% |
| | Non-vegetarian | 141 | 69% |
| | Flexitarian | 30 | 15% |
| | Pescatarian | 2 | 1% |
| | Others | 16 | 8% |
| | Prefer not to say | 1 | 0% |

**Table 1.** *Cont.*

| Variable | Category | Number | Percentage |
|---|---|---|---|
| Frequency of meat consumption | Always | 44 | 21.36% |
| | Most of the time | 100 | 48.54% |
| | Sometimes | 34 | 16.50% |
| | Rarely | 12 | 5.83% |
| | Never | 16 | 7.77% |
| N = 206 | | | |

### 3.1.2. Validity Testing of Means, KMO and Cronbach's Alpha

Means

The mean and standard deviations of variables in the conceptual model are shown in Table 2. The mean values of most variables are in the range of between 2 and 3 on a 5-point scale, which correspond to poor perceptions and purchase intentions. Poor perceptions and purchase intentions have been attributed to consumer reluctance towards novel food products, especially if it is genetically modified due to techno scepticism, food neophobia [10,16], and disgust due to unnaturalness [11,40].

**Table 2.** Variables extracted from the conceptual framework along with their mean, SD, KMO and Cronbach's alpha values.

| Factors | Mean * | SD | Cronbach's Alpha | KMO ** |
|---|---|---|---|---|
| Environment and Sustainability | 3.18 | 1.18 | 0.861 | 0.854 |
| Health & Safety | 2.72 | 0.95 | 0.573 | 0.714 |
| Cultural Belief | 2.05 | 1.08 | 0.622 | 0.749 |
| Current Purchase Behaviour and Consumption | 2.39 | 1.14 | 0.692 | 0.551 |
| IVM Purchase | 2.93 | 1.01 | 0.634 | 0.603 |

* The mean scores as seen in the table are based on the 5-point Likert scale; ** Kaiser Meyer Olkin measures the sampling adequacy.

Reliability

The factors of the conceptual model were measured for reliability using the Cronbach's alpha value, which tested the subscales for coherent dimensions [41]. Alpha values between 0.5 to 0.7 and 0.7 to 1 are considered as moderately reliable and highly reliable, respectively [42]. Values in this study ranged between 0.57 and 0.86 (Table 2), which indicates an exceptionally good consistency [40,43].

Sampling Adequacy

The Kaiser Meyer Olkin (KMO) values were used as a measure of sample adequacy for each factor in the conceptual model [44]. Research showed that a KMO value greater than 0.5 was considered a good sample size [45]. Results from this study (Table 2) showed KMO values between 0.50 to 0.90, meaning that it was acceptable [37].

### 3.1.3. Partial Least Squares Path Modelling (PLS-PM): Structural Model Assessment

PLS-PM analysis was performed to examine the relationship of variables investigated in this study. The results (Table 3) showed that all variables had positive β-coefficients. Furthermore, all variables in the conceptual model had significant *p*-values ($p < 0.05$), which indicated that all variables had a significant effect on purchase and consumption behaviour.

**Table 3.** The β-coefficients and *p*-values of all variables in the conceptual framework showing the effect of all variables on purchase and consumption behaviour.

| Variables | β-Coefficients | *p*-Value |
|---|---|---|
| Current purchase behaviour and consumption → IVM purchase | 0.097 | Pr > \|t\| = 0.034 |
| Cultural beliefs → IVM purchase | 0.516 | Pr > \|t\| < 0.001 |
| Health and Safety → IVM purchase | 0.316 | Pr > \|t\| < 0.001 |
| Environment and sustainability → IVM purchase | 0.100 | Pr > \|t\| = 0.036 |

3.1.4. Analysis of TPB Factors in the Conceptual Model

IVM Purchase and Consumption Behaviour

In this study, the IVM purchase and consumption behaviour variable measure consumers' willingness to try (WTT), willingness to buy (WTB), and willingness to eat (WTE). Both IVM purchase and consumption behaviour were influenced by their views on IVM and other variables (such as environment and sustainability, health and safety, cultural beliefs and current purchase and consumption behaviour). As a result, these variables had a significant and robust relationship.

Typically, most studies examine the effect of perception on IVM purchase and consumption, but different terms were used (like absolute opposition [9] and acceptance [10,11]), and different aspects of purchase and consumption behaviour were investigated (like willingness to eat (WTE) [8,9] and willingness to purchase (WTP) [19]). A similar approach was observed in other studies [8,10,11,16,19,46].

Table 4 shows that 44% of New Zealanders were neutral about trying IVM over traditional meat (Mean = 3.31, SD = 1.05) even though IVM had a health-friendly nutrition profile (Mean = 3.42, SD = 1.05). In addition, 32% of New Zealand consumers were neither willing nor unwilling about buying IVM over traditional meat (Mean = 3.35, SD = 0.98). As seen in Table 4, the means for most questions were close to 3 on a 5-point scale, which implies that consumers had neutral opinions. Neutrality in opinion may be due to the nascent IVM technology itself. Since IVM technology is relatively new and unknown to consumers, neutral opinions are given.

Results showed that 50% of NZ consumers would purchase IVM regularly (Mean = 2.97, SD = 1.04) compared to traditional meat. Furthermore, 53% of NZ consumers would not buy IVM (Mean = 2.4, SD = 1.04) if more highly priced than traditional meat. From the results in Table 4, it is evident that New Zealand consumers have an interest in engaging with IVM, and price is one of the key barriers for their purchase behaviour. New Zealand consumers are one of the highest meat consumers in the world [47,48]. As a result, their diet is mainly meat-based and, in such cases, there will be higher meat attachment and higher inclination to try meat alternatives [10]. Thus, New Zealanders were equally interested in trying or buying IVM.

Most New Zealand consumers (36%) were unaware of IVM and only 23% were slightly aware of IVM (Mean = 2.27, SD = 1.24), indicating that the familiarity of IVM among consumers was very low. Familiarity is created when there is media exposure to IVM technology. Unfortunately, IVM awareness in NZ is not as high compared to European (UK, Belgium, France, Italy and Netherland) and other Western countries (USA and Canada). Thus, New Zealanders were less interested in trying or buying IVM. This may account for poor IVM purchase and consumption intentions amongst NZ consumers. Perception of IVM has been shown to improve when consumers (*n* = 525) were provided with more information on IVM, and as a result, even increased purchase and consumption behaviour (willingness to try-WTT = 54% and willingness to buy-WTB = 44.2%) [19].

**Table 4.** Means of survey questions on behavioural intentions towards IVM purchase and consumption.

| Variables. | Question | Mean | SD | Distribution: 1–2 Score | Distribution: 3 Score | Distribution: 4–5 Score | Scale Anchors |
|---|---|---|---|---|---|---|---|
| Willingness to try (WTT) | Do you think you would try in-vitro meat (IVM) for its nutritional profile? | 3.42 | ±1.05 | 13% | 39% | 48% | |
| | Do you think you will try it over traditional meat? | 3.31 | ±1.05 | 15% | 44% | 41% | |
| Willingness to buy/purchase (WTB/WTP) | Do you think you would buy in-vitro meat (IVM) over traditional meat? | 3.35 | ±0.98 | 19% | 32% | 50% | |
| | Do you think you would buy in-vitro meat (IVM) if it is affordable | 3.53 | ±1.08 | 13% | 31% | 56% | Definitely not–definitely yes |
| | Do you think you would buy in-vitro meat (IVM) regularly | 2.97 | ±1.04 | 32% | 35% | 33% | |
| | Do you think you would buy in-vitro meat (IVM) if it is labelled as guilt-free meat? | 3.64 | ±1.20 | 12% | 30% | 58% | |
| | Do you think you would buy in-vitro meat (IVM) if it would be cheaper than conventional meat? | 3.4 | ±0.97 | 17% | 34% | 49% | |
| | Do you think you would buy in-vitro meat (IVM) over conventional meat even though it is expensive? | 2.4 | ±1.04 | 53% | 35% | 11% | |

Environment and Sustainability

Environmental attitude towards IVM depends on factors related to general environmental and animal welfare awareness, as well as consumers' openness towards sustainable alternatives. Overall effect of the environmental and sustainability variable on IVM purchase was significant (Table 2). There was a positive correlation between the variables ($\beta$ = 0.100). In other words, a one unit increase in consumers' perceptions on environment and sustainability will lead to increase in IVM purchase intentions among consumers. Hence, the environment and sustainability variable was a significant factor that influenced IVM purchase intentions in this study. The means for questions under the environment and sustainability variable (Table 5) showed varied results. It is evident from the results that although 72% of NZ consumers are environmentally conscious (Mean = 3.83, SD = 1.01) and 70% are pro-animal welfare (Mean = 3.84, SD = 1.11), not many consumers are aware of the negative repercussions of the traditional meat industry (Mean = 2.94, SD = 1.29). As a result, most consumers do not understand the importance of sustainable alternatives and have decreased purchase intentions towards IVM, as seen in Table 5.

On the contrary, a study [9] showed that US consumers (*n* = 673, Mean = 1.97, 1-much more–5-much less) who were environmentally conscious agreed that IVM was an environmentally friendly alternative, and subsequently resulted in higher purchase and consumption intentions. Similarly another study [19], showed that environmentally conscious Italian consumers (*n* = 525) perceived IVM favourably that resulted in higher purchase intentions (44.2% said yes to willingness to buy IVM).

**Table 5.** Means of survey questions on environmental and animal welfare awareness that indirectly influence consumer perceptions of IVM.

| Variables | Questions | Mean | SD | Distribution: 1–2 Score | Distribution: 3 Score | Distribution: 4–5 Score | Scale Anchors |
|---|---|---|---|---|---|---|---|
| Environment and Sustainability | How important is the environment to you? | 3.83 | ±1.01 | 13% | 15% | 72% | Not at all important–extremely important |
| | Are you aware of the negative environmental effects of the conventional meat industry? | 2.94 | ±1.29 | 39% | 31% | 31% | Not at all aware–extremely aware |
| | Do you think that the traditional meat industry contributes to global issues such as greenhouse gas emission and changes in climate? | 3.62 | ±1.14 | 14% | 30% | 56% | Definitely not–definitely yes |
| | Are you aware of the fact that the conventional meat industry has a higher carbon footprint compared to other meat alternatives? | 2.99 | ±1.31 | 37% | 29% | 34% | Not at all aware–extremely aware |
| | Are you aware of the terms such as sustainability and a sustainable environment? | 3.46 | ±1.22 | 22% | 24% | 54% | |
| | Do you believe in animal welfare? | 3.84 | ±1.11 | 12% | 18% | 70% | Definitely not–definitely yes |
| | Are you familiar of sustainable meat alternatives? | 2.64 | ±1.14 | 51% | 25% | 24% | Not at all familiar–extremely familiar |
| | Do you think you are open to technologies related to the food industry? | 3.59 | ±1.11 | 16% | 22% | 61% | Definitely not–definitely yes |
| | Are you familiar with meat analogues such as plant-based protein, insect protein, in-vitro meat (IVM), vegan fish and fishless seafood? | 2.66 | ±1.19 | 46% | 29% | 24% | Not at all familiar–extremely familiar |
| | Are you familiar with technologies such as cellular agriculture/In-vitro meat (IVM) technology and tissue engineering? | 2.21 | ±1.18 | 60% | 25% | 15% | |

The environmental and sustainability variable also measures consumer's general awareness and their willingness towards sustainable alternative consumption, such as plant-based meat and cell-based meats. The underlying question here was to check for food neophobia among consumers towards novel food products [9,10,49,50]. The means of questions in Table 6 showed that New Zealand consumers are not entirely open to sustainable alternatives. A study with US consumers [9,51] reported that consumers had moral absolutism (Moral absolutism is the belief that the morality or immorality of an action can be judged according to fixed standards of right and wrong) and disgust towards genetically modified (GM) foods. In addition, moral absolutism was found to be more evident among US consumers compared to European consumers [51]. The reduced openness to sustainable alternatives in this study could also be linked to subtle absolutism. Furthermore, reduced openness towards sustainable alternatives can also be due to eating behaviour. For example, consumers with high meat consumption have minimal interest in engaging with sustainable alternatives [14], as is the case with New Zealand consumers. Additionally, openness to sustainable alternatives, and consumer reluctance towards IVM

is also due to other factors like unnaturalness [8], disgust, fear of perceived risks [13] and food neophobia [9,10].

**Table 6.** Means of survey questions on the health and safety profile of IVM that indirectly influence IVM purchase and consumption behaviour.

| Variable | Questions | Mean | SD | Distribution: 1–2 Score | Distribution: 3 Score | Distribution: 4–5 Score | Scale Anchors |
|---|---|---|---|---|---|---|---|
| Health and Safety | Do you think in-vitro meat (IVM) should be produced by government-approved agencies? | 2.43 | ±0.99 | 52% | 38% | 10% | Definitely not–definitely yes |
| | Do you think in-vitro meat (IVM) has any health and safety concerns? | 3.27 | ±0.94 | 17% | 45% | 39% | |
| | Do you think in-vitro meat (IVM) is likely to cause any disease? | 2.47 | ±1.01 | 53% | 35% | 12% | |
| | Do you think in-vitro meat (IVM) is cancerous as it involves stem cells? | 2.86 | ±0.89 | 25% | 59% | 17% | |
| | Do you think in-vitro meat (IVM) will have any food safety risk? | 2.54 | ±0.88 | 47% | 41% | 12% | |

The results in Table 5 show that consumers were not familiar with sustainable alternatives available in the market. In addition, NZ consumers were less familiar with IVM, cell-based meats, and other plant-based protein. Similarly, only 13% (*n* = 180) of Belgian consumers [22] and 14% (*n* = 1296) Netherlands consumers were familiar with IVM [17]. In addition, US consumers (57.3%), Chinese consumers (35.5%) and Indian consumers (25.5%) were not at all familiar with IVM (Bryant et al., 2019c). Generally, lack of familiarity with IVM among consumers is one of the primary concerns for poor purchase intentions, as previously explained.

Health and Safety

Consumer opinions on the health and safety aspects of IVM mainly focus on food safety and perceived health concerns, such as cancer and other food-related diseases. These opinions collectively form the health and safety variable, which indirectly affects IVM purchase intentions. In our study, health and safety variables are vital as they measure consumer concerns, which influence consumer perceptions of IVM. Consumers are wary about the perceived health and safety risks of IVM [13] due to unnaturalness, scientific distrust and unfamiliarity [13].

Results in Table 6 show that IVM health and safety (Mean = 2.72, SD = 0.95) had a significant effect on IVM purchase and consumption behaviour. There was a positive correlation (β = 0.316) between the factors, which suggest that perceived health and safety concerns on IVM are likely to influence IVM purchase intentions. Findings on IVM in terms of health and safety perception have also been reported in other studies [8,14,16,19,40]. Some studies showed that consumers who were not in favour of IVM due to health and safety reasons led to consumers having decreased purchase intentions [12,14].

Results in Table 6 showed that 52% of New Zealand consumers would probably not buy IVM (Mean = 2.43, SD = 0.99) and 38% were neutral about buying IVM, even if it was manufactured by government-approved agencies. Furthermore, 39% of New Zealand consumers had health and safety concerns about IVM, whereas 45% of consumers were neutral about IVM health and safety (Mean = 3.27, SD = 0.94). On the other hand, 12% of NZ consumers perceived IVM as probably having food safety risk and 41% of consumer had neutral opinions on IVM food safety risk (Mean = 2.54, SD = 0.88). These findings could be due to the nature of IVM, as it is an artificial/man-made food product using animal cells. Health and safety concerns were one of the common reasons for rejection of IVM, and this concern is widely reported in previous IVM literature [8,11,13,16,19,52]. Health and safety concerns appear to be related to the perceived unnaturalness of tissue engineering. Due to the involvement of genetic modification, IVM is susceptible to disgust,

concerns on unnaturalness and fear of perceived risks [13]. IVM is often perceived as being 'unnatural' as it is often confused with genetically modified foods due to both involving biotechnology methods for production. IVM is in fact non-GMO when produced from unmodified cells extracted from animals through biopsy. However, genetic modification might be applied to increase the production efficiency of IVM in the future.

Religious and Cultural Beliefs

The religious and cultural beliefs variable measures the IVM purchase intention based on the consumers' cultural identity and beliefs. The underlying question here was to check for cultural implications on IVM as a sustainable meat alternative. The effect of religious and cultural beliefs (Mean = 2.05, SD = 1.08) on IVM purchase and consumption behaviour was significant. There was a positive correlation between the variables ($\beta$ = 0.516), which suggest that these beliefs are likely to influence IVM purchase intentions.

The results in Table 7 showed that NZ consumers (Mean = 2.4, SD = 1.08) would not consume IVM, even if their religious beliefs permitted (Mean = 2.4, SD = 1.09), their religious leaders promoted IVM (Mean = 2.16, SD = 1.08), or if IVM was available as halal or kosher meat (Mean = 1.89, SD = 1.05). It is most likely that the majority of NZ consumers are giving negative answers to indicate that this is not a relevant consideration to them. Only 43.69% of the respondents identified themselves as being religious in this study (Table 1).

**Table 7.** Means of survey questions on consumers' religious and cultural beliefs that indirectly influence IVM purchase and consumption behaviour.

| Variable | Question | Mean | SD | Distribution: 1–2 Score | Distribution: 3 Score | Distribution: 4–5 Score | Scale Anchors |
|---|---|---|---|---|---|---|---|
| Cultural beliefs | Do you think you would likely consume in-vitro meat (IVM) if your religious beliefs permitted? | 2.4 | ±1.09 | 44% | 38% | 18% | Definitely not–definitely yes |
| | Do you think you would eat in-vitro meat (IVM) if the religious leaders informed you? | 2.16 | ±1.08 | 25% | 54% | 21% | |
| | In some religions, the intellectual level of the animal to be slaughtered for meat purposes are considered, such as "only those animals with lower intellectual capacity and pain sensation are to be slaughtered" whereas, in-vitro meat (IVM) does not require any animal slaughter at all. Do you think in-vitro meat (IVM) is a better meat alternative than conventional meat? | 1.73 | ±1.09 | 17% | 42% | 41% | |
| | Do you think you would opt for in-vitro meat (IVM) if there will be halal [1] or kosher [2] options available? | 1.89 | ±1.05 | 18% | 53% | 28% | |

[1] Halal—refers to what is permissible or lawful in traditional Islamic law; [2] Kosher—Kosher foods are those that conform to the Jewish dietary regulations of kashrut (dietary law).

Current Purchase Behaviour and Consumption

The current purchase behaviour and consumption variable measures the IVM purchase intentions based on consumers current purchase behaviour and meat-eating behaviour. The underlying idea here is to test the effect of current eating and purchase behaviour on IVM purchase and consumption behaviour. The effect of current purchase behaviour and consumption (Mean = 2.39, SD = 1.14) on IVM purchase and consumption behaviour was significant. There was a positive correlation between the variables ($\beta$ = 0.097), which suggests that current purchase behaviour and consumption are likely to influence IVM purchase intentions.

Results in Table 8 show that 70% of NZ consumers had higher meat intake ($n$ = 206, Mean = 3.69, SD = 1.10), with poultry (71%) being the most preferred choice of meat. These results showed that NZ consumers had a higher meat attachment and would not give up meat for the welfare of animals (Mean = 2.78, SD = 1.27). However, 32% of NZ consumers were uncertain about buying traditional meat due to its carbon footprint (Mean = 3.14, SD = 1.18) but were willing to buy plant-based meat alternatives compared to traditional meat products (Mean = 4.08, SD = 1.18), mainly due to its nutritional profile. These findings demonstrated that although NZ consumers are fond of meat, they were willing to engage with plant-based meat alternatives but were reluctant to try IVM. The main barriers towards IVM among NZ consumers can be due to several reasons, such as the nature of IVM, differences in eating habits, higher meat attachment and general food neophobia as previously discussed.

**Table 8.** Means of survey questions on consumers' current purchase behaviour and consumption that indirectly influence IVM purchase and consumption behaviour.

| Variables | Question | Mean | SD | Distribution: 1–2 Score | | Distribution: 3 Score | | Distribution: 4–5 Score | | Scale Anchors (or Selection) |
|---|---|---|---|---|---|---|---|---|---|---|
| Current purchase behaviour and consumption | How often do you eat meat? | 3.69 | ±1.11 | 14% | | 17% | | 70% | | Never–Always |
| | Which type of meat do you prefer the most? (Check all that applies) | - | - | 17% | 13% | 58% | 10% | 2% | | (a) Poultry (b) Pork (c) Beef (d) Fish and other seafood (e) Others |
| | What does your meat intake look like? | 3.49 | ±0.9 | 3% | 6% | 30% | 60% | 1% | | (a) Once a month (b) Once a fortnight (c) Once a week (d) Every meal (e) Never |
| | When do you usually eat meat? When I am ...? (Check all that applies) | - | - | 20% | 3% | 48% | 14% | 15% | | (a) Hungry (b) Bored (c) Crave (d) Sad (e) Happy |
| | Do you think you will still buy conventional meat although it has a higher carbon footprint? | 3.14 | ±1.18 | 23% | | 32% | | 45% | | Definitely not–definitely yes |
| | Do you think you would give up meat for the sake of animals? | 2.78 | ±1.27 | 43% | | 30% | | 27% | | |
| | Do you think you would buy plant-based products if it had a better health star rating compared to meat products? | 4.08 | ±1.18 | 69% | | 10% | | 15% | | |

## 4. Conclusions

This study sets out to understand the perceptions of IVM by New Zealand consumers, particularly with respect to IVM purchase and consumption behaviour. The general results indicated that if the overall perception of IVM improved, then the purchase likelihood and consumption behaviour would improve as well. Although perception plays an important role in purchase intention, other variables such as consumers' current purchase and consumption behaviour, environment and sustainability awareness and consumers opinions on health and safety of IVM, also influence consumers' purchase intentions. However, the results also revealed that consumers' religious and cultural beliefs had minimal influence on IVM purchase likelihood. It is important to note that there was no pre-screen or checks

for the consumers' religious beliefs. Hence, in this instance, the concept of halal and kosher may not be relevant for non-Muslim or non-Jewish groups.

Results in this study indicated that most New Zealand consumer have minimal awareness of IVM, as a plurality of consumers were neutral in terms of engaging with IVM. In addition, New Zealand consumers are hesitant to engage with IVM due to lack of familiarity and thus NZ consumers need to be educated about IVM and its potential benefits. Awareness of IVM can bring about a change in its perception, and in some cases even lead to consumer acceptance.

These findings add to a growing body of literature on the perception of IVM among New Zealanders. However, there may be some possible limitations in this study. The first limitation of this study is the use of convenience sampling, which may have resulted in partly biased findings. Thus, it would be beneficial to understand the perception of the general population in future studies. Furthermore, this study did not test consumers for food neophobia. Further work can investigate whether consumer reluctance towards IVM is due to food neophobia or if it is merely due to extensive meat intake. Additionally, it would be interesting for future studies to provide a blinded (without knowing its origin) meat sample for consumers to further understand consumers' perception and preferences.

**Author Contributions:** Conceptualization, M.M.M., N.H., A.S.; Methodology, M.M.M., N.H., K.K., Y.L., A.S.; Software, N.H., K.K.; Validation, M.M.M., Y.L.; Formal analysis, M.M.M., N.H., K.K., Y.L.; Investigation, M.M.M., Y.L.; Resources, N.H., A.S.; Data Curation, M.M.M., K.K.; Writing—Original Draft, M.M.M., N.H., K.K., Y.L., A.S.; Writing—Review & Editing, M.M.M., N.H., K.K., Y.L., A.S.; Visualization, M.M.M., K.K.; Supervision, N.H., K.K., A.S.; Project administration, N.H., A.S.; Funding acquisition, N.H., A.S. All authors have read and agreed to the published version of the manuscript.

**Funding:** This research received no external funding.

**Institutional Review Board Statement:** The study was conducted according to the guidelines of the Declaration of Helsinki, and approved by the Auckland University of Technology Ethics Committee (AUTEC 19/68, 12 April 2019).

**Informed Consent Statement:** Informed consent was obtained from all subjects involved in the study.

**Conflicts of Interest:** The authors declare no conflict of interest.

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
