# Peer review of "Consumers’ Perception of In-Vitro Meat in New Zealand Using the Theory of Planned Behaviour Model"

_sustainability, doi:10.3390/su13137430_

Round 1

Reviewer 1 Report

Abstract

Your abstract is very thorough, and could be a bit shorter - usually these don’t need to go into limitations, future research etc.

  1. Introduction

For a more recent review of IVM overall, see Post et al. (2020) - https://www.nature.com/articles/s43016-020-0112-z 

Line 66: It is not necessarily straightforward to compare percentages from these different studies, since each study had different descriptions of IVM, different question and answer options, and different sampling procedures - the differences are more likely due to these factors than differences between countries, unless the comparison is being made within the same study or using the same methods. There is one (non-peer reviewed) survey which seemed to indicate higher acceptance in the US vs. the UK - https://www.datasmoothie.com/@surveygoo/nearly-one-in-three-consumers-willing-to-eat-lab-g/ - but the sampling here is not ideal. I have a study under review which shows very similar levels of acceptance in the UK and the US - you can cite (Szejda, Bryant & Urbanovich, forthcoming) to note that these were similar. I recommend re-writing this paragraph.

Line 90: You can cite Bryant & Barnett (2020) for a recent review of relevant literature - https://www.mdpi.com/2076-3417/10/15/5201/htm

Line 91: Citation needed comparing NZ meat consumption to other countries

Line 111: While it is true that left-leaning people tend to have higher acceptance of IVM, it is not that these people are necessarily environmentalists or welfare activists. In fact, research shows that IVM is most appealing to people with the highest meat consumption.

Line 127: I don’t think we can say that IVM requires advancements in terms of taste and/or safety - it is just that consumers think these might be problems. They have not actually got experience of the product in these studies.

Line 140: No need to state the null hypothesis.

  1. Materials and methods

I would like to see more about how the participants were recruited. In person - where were they? A university campus, a public space? And online - which Facebook or LinkedIn groups were used? This information is necessary to assess potential bias.

Also, please provide the descriptions of IVM participants read.

Please provide a description of PLS-PM, since most readers will be unfamiliar with this method.

  1. Results and discussion

Citation needed for interpreting Cronbach’s alpha values. Generally 0.57 is rather low, which is worth commenting on (though it is still acceptable).

Table 4: some of the scale anchors seem to be incorrect - e.g. the question is ‘Do you think…’ and the answer options are ‘Agee-disagree’. 

Line 302: Table 4 actually does not contain any percentages, though these would be useful to add. As well as the mean values on the Likert scale, you should report the percentage who indicated a negative (1-2) neutral (3) or positive (4-5) response.

Line 313: How can you say it is evident that NZ consumers have minimal interest in engaging with IVM when the values are overall quite positive? The mean values are generally above the mid-point, and you have a high percentage saying they would eat it. In fact, you have only reported percentages for the neutral or negative groups, but not for those who said they would eat it. Please add these figures to Table 4 to amend this bias.

Line 320: Again, this is not the case. Higher meat attachment generally predicts higher CM purchase intentions, not lower - https://www.frontiersin.org/articles/10.3389/fsufs.2019.00011/full 

Line 341: This Hocquette study is not a good source because the sample is extremely non-representative (scientists and meat industry workers!) This more recent study from France showed that 44% in France would try CM - https://www.mdpi.com/2304-8158/9/9/1152/htm 

Line 347: The Slade finding was not 0.126 on a 5-point scale (these scales are typically 1-5) but rather that 12.6% of respondents selected IVM when it was available.

Table 5: Again, it would be very helpful to show the percentage answering positive, neutral, or negative to each statement, as well as the means and SDs. That way, you can say things like ‘most people agreed’

Line 396: 35% and 25% are not a majority

P15: Better citation than Wikipedia needed… Stanford Encyclopedia of Philosophy perhaps?

Line 410: Table 6 does not show any relationship with consumption behaviour, only mean values. Again, you could add percentages to this Table.

Line 417: Again, Hocquette’s sample was not at all representative of French consumers

Line 420: Table 6 does not show percentages

Line 433: Tissue engineering is not genetic modification. IVM is not genetically modified.

Table 7: Again, no percentages are shown here - you should add them. 

All of the measures labelled ‘cultural beliefs’ are in fact religious beliefs. ‘Cultural beliefs’ could be broader, encompassing beliefs about the place of ranching in culture etc. I suggest calling these variables ‘religious beliefs’

You may also want to restrict this analysis to people who follow the relevant religions, or else the results will be very watered-down. Of course, Halal and Kosher are not relevant for anyone who is not Muslim or Jewish, but they may be relevant to those groups. The conclusion that ‘purchase intention is not influenced by cultural beliefs’ is probably following from the fact that these questions were not relevant for many consumers.

Table 8: Some of these variables do not make sense to report as means, e.g. rows 2 and 4. Instead, you should report the percentage who selected each option.

Again, add percentages - your explanation says that Table 8 shows some percentage, but it does not. 

Your conclusions in this section do not clearly follow from the results in Table 8. You haven’t reported asking anything about plant-based alternatives.

P19: This section is cut off.

  1. Conclusions

You say that consumers are mostly neutral, but less than 50% were neutral. You could say ‘a plurality were neutral’ (e.g. more than any other group).

The conclusion can be much improved by better analysis (see my suggestions above).

Author Response

Please see attached for review response

Reviewer 2 Report

The article is well written and defined. Even so, before raising the hypotheses, the bases of the same should be explained more fully. The article is very interesting, but there are questions that could have been asked blindly, that is, which of these meats would you choose, without previously knowing its origin, although it is probably not the bottom line. I find it particularly interesting because over time it will be one of the ways to consume meat and where it comes from.

Author Response

Please see attached for review response

Round 2

Reviewer 1 Report

Line 48: I appreciate the addition of an explanation of why different surveys may not be comparable, but this paragraph is still phrased as Europe having higher acceptance than the US, and this is not the case. This new study compares cultured meat acceptance in the US/UK, finding that they are similar: https://www.mdpi.com/2304-8158/10/5/1050 

Line 141: Please provide the exact information given to participants. You can include this as an Appendix if necessary. 

Line 348: I think it would be clearer to say ‘Health and safety concerns appear to be related to the perceived unnaturalnes of tissue engineering.’

Line 365: I think this interpretation is incorrect - this figure is quite a lot lower than the general willingness to try. It seems unlikely that people would be LESS likely to eat it if their religious leaders allowed it - most likely, the majority of respondents are giving negative answers to indicate that this is not a relevant consideration to them. This seems to be the only reasonable interpretation for why there would be fewer people saying they would eat IVM if their religious leader endorsed it compared to people saying they would eat IVM in general. 

These religious analyses are potentially useful, but only in the relevant religious groups. If the authors have the space or inclination, I would like to see the importance of Kosher/Halal for respondents who actually were Jewish/Muslim respectively - otherwise, these factors have not been properly assessed and I suggest leaivng them out. The current conclusions about the implications of this religious data are incorrect.

Table 8: The row ‘What does your meat intake look like’ has 5 responses but only 3 columns - make this consistent with the other 5-choice rows? Also, the percentages in the final row are incorrect (they do not correspond to the mean presented and do not add to 100%)

Line 391: Again, you say here that consumers were reluctant to try IVM, but that does not fit with your findings. 53% would buy if it was affordable, 50% would buy over traditional meat - these are encouraging numbers, it is not clear why the authors are presenting them as ‘consumers are hesitant’?

Author Response

Please see attached revision

Round 3

Reviewer 1 Report

The authors have made some improvements to the paper, but I still have some comments.

Line 48: I'm afraid I still find this paragraph misleading. You cannot say that acceptance is overall higher in Europe compared to the USA. The only studies we have making this comparison suggest there is no difference, or the reverse is true. Please re-write this paragraph so that it does not imply acceptance is lower in the US.

Line 151: Please use quote marks to indicate this passage is quoted.

Line 405: Please remove the claim that IVM is genetically modified. This is not necessarily the case.
